# The Constellation of Risk Factors and Paraneoplastic Syndromes in Cholangiocarcinoma: Integrating the Endocrine Panel Amid Tumour-Related Biology (A Narrative Review)

**DOI:** 10.3390/biology13090662

**Published:** 2024-08-26

**Authors:** Mihai-Lucian Ciobica, Bianca-Andreea Sandulescu, Liana-Maria Chicea, Mihaela Iordache, Maria-Laura Groseanu, Mara Carsote, Claudiu Nistor, Ana-Maria Radu

**Affiliations:** 1Department of Internal Medicine and Gastroenterology, “Carol Davila” University of Medicine and Pharmacy, 020021 Bucharest, Romania; lucian.ciobica@umfcd.ro (M.-L.C.); bianca-andreea.sandulescu@drd.umfcd.ro (B.-A.S.); ana-maria.radu@rez.umfcd.ro (A.-M.R.); 2Department of Internal Medicine I and Rheumatology, “Dr. Carol Davila” Central Military University Emergency Hospital, 010825 Bucharest, Romania; 3PhD Doctoral School of “Carol Davila” University of Medicine and Pharmacy, 020021 Bucharest, Romania; 4Clinical Medical Department, University “Lucian Blaga” Sibiu, 550024 Sibiu, Romania; liana.chicea@ulbsibiu.ro; 51st Internal Medicine Department, “Dr. Carol Davila” Central Military University Emergency Hospital, 010825 Bucharest, Romania; mihaelaiordache2005@yahoo.com; 6Internal Medicine and Rheumatology Department, “Carol Davila” University of Medicine and Pharmacy, 020021 Bucharest, Romania; maria.groseanu@umfcd.ro; 7Department of Endocrinology, “Carol Davila” University of Medicine and Pharmacy, 050474 Bucharest, Romania; 8Department of Clinical Endocrinology V, C.I. Parhon National Institute of Endocrinology, 011863 Bucharest, Romania; 9Department 4–Cardio-Thoracic Pathology, Thoracic Surgery II Discipline, “Carol Davila” University of Medicine and Pharmacy, 050474 Bucharest, Romania; 10Thoracic Surgery Department, “Dr. Carol Davila” Central Emergency University Military Hospital, 010825 Bucharest, Romania

**Keywords:** cholangiocarcinoma, endocrine, paraneoplastic syndrome, risk factor, calcium, PTHrP, tumour biology

## Abstract

**Simple Summary:**

Cholangiocarcinomas, malignant tumours originating from the biliary epithelium, have had an increasing incidence during recent decades; thus, awareness represents the key operating factor nowadays. Our purpose was to overview the tumour field following the constellation of the risk factors and the paraneoplastic syndromes, emphasizing the endocrine features amid the entire multidisciplinary panel. The hormonal burden is reflected by complex interplays such as the galanin system, exposure to sex hormone therapy, interferences with vitamin D system, etc., while humoral hypercalcaemia of malignancy, although rare, has been reported due to parathyroid hormone-related protein over-production in addition to other elements such a fibroblast growth factor receptor-positive or even procalcitonin-positive tumour. The level of endocrine evidence across clinical trials remains far from generous. Further applications such as emergent hormonal biomarkers are expected for daily practice.

**Abstract:**

Cholangiocarcinomas (CCAs), a heterogeneous group of challenging malignant tumours which originate from the biliary epithelium, are associated with an alarming increasing incidence during recent decades that varies between different regions of the globe. Thus, awareness represents the key operating factor. Our purpose was to overview the field of CCAs following a double perspective: the constellation of the risk factors, and the presence of the paraneoplastic syndromes, emphasizing the endocrine features amid the entire multidisciplinary panel. This is a narrative review. A PubMed-based search of English-language original articles offered the basis of this comprehensive approach. Multiple risk factors underlying different levels of statistical evidence have been listed such as chronic biliary diseases and liver conditions, inflammatory bowel disease, parasitic infections (e.g., *Opisthorchis viverrini*, *Clonorchis sinensis*), lifestyle influence (e.g., alcohol, smoking), environmental exposure (e.g., thorotrast, asbestos), and certain genetic and epigenetic interplays. With regard to the endocrine panel, a heterogeneous spectrum should be taken into consideration: non-alcoholic fatty liver disease, obesity, type 2 diabetes mellitus, and potential connections with vitamin D status, glucagon-like peptide 1 receptor, or the galanin system, respectively, with exposure to sex hormone therapy. Amid the numerous dermatologic, hematologic, renal, and neurologic paraneoplastic manifestations in CCAs, the endocrine panel is less described. Humoral hypercalcaemia of malignancy stands as the most frequent humoral paraneoplastic syndrome in CCAs, despite being exceptional when compared to other paraneoplastic (non-endocrine) manifestations and to its reported frequency in other (non-CCAs) cancers (it accompanies 20–30% of all cancers). It represents a poor prognosis marker in CCA; it may be episodic once the tumour relapses. In addition to the therapy that targets the originating malignancy, hypercalcaemia requires the administration of bisphosphonates (e.g., intravenous zoledronic acid) or denosumab. Early detection firstly helps the general wellbeing of a patient due to a prompt medical control of high serum calcium and it also provides a fine biomarker of disease status in selected cases that harbour the capacity of PTHrP secretion. The exact molecular biology and genetic configuration of CCAs that display such endocrine traits is still an open matter, but humoral hypercalcaemia adds to the overall disease burden.

## 1. Introduction

Cholangiocarcinomas (CCAs), a heterogeneous group of challenging malignant tumours which originate from the biliary epithelium, are anatomically classified into intrahepatic (iCCAs), perihilar (pCCAs), and distal (dCCAs), the last two also being referred to as extrahepatic (eCCAs) [1,2]. iCCAs are located proximally to the second-order bile ducts and comprise 10–20% of all CCAs [1,3]. pCCA, also known as Klatskin tumour, is the most prevalent type of CCA (50–60%) and arises above the insertion of the cystic duct into the common bile duct [1,3,4]. dCCA can be located anywhere between the cystic duct and the ampulla of Vater and represents 20–30% of all CCAs [1,3,4]. Although iCCA is the least prevalent type of cholangiocarcinoma, it is the second most common type of hepatic malignancy after hepatocellular carcinoma [1,3]. Histologically, most CCAs are adenocarcinomas, with other subtypes, such as clear cell carcinomas or adenosquamous carcinomas, being very rare [1,2,3,4].

An alarming increasing incidence of this group of ailments has been reported during recent decades. It varies between different regions of the globe, with, for instance, between 0.3 and 6 cases per 100,000 inhabitants per year in Western countries and >6 cases per 100,000 inhabitants in Eastern Asia, especially Thailand, South Korea, and China [5,6]. Moreover, the disease burden, including a high mortality rate, requires a multidisciplinary awareness from the early stages for a better outcome. In Europe, data collected between 2008 and 2018 showed that the highest mortality rates were registered in Malta, Ireland, and Spain for male patients and in the UK and Switzerland for female patients, while Romania, Hungary, and Poland reported the lowest rates for both sexes, with a general increase in iCCA-related mortality for both sexes in all countries over the past decade [7,8]. This variation could be explained by geographical differences in associated risk factors as well as genetic determinants [1,2]. 

Our purpose was to overview the field of CCAs following a double perspective: the constellation of the risk factors, and the paraneoplastic syndromes, emphasizing the endocrine features amid the entire multidisciplinary panel. This is a narrative review. A PubMed-based search of English-language original articles offered the basis of this comprehensive multidisciplinary approach. The search was carried out from inception until June 2024, and most papers were published within the last decade.

## 2. CCA-Associated Risk Factors and Potential Contributors

Multiple risk factors have been reported in relationship with the group of CCAs. Some of them imply strong statistical evidence with the entire category of neoplasia while others have been related with a specific CCA subtype, but chronic inflammation of the biliary epithelium and bile stasis seem to be common characteristics amid the constellation of the multidisciplinary risk factors that have been identified so far [1,4,6]. Despite advancements in understanding the aetiology of this disease, around 50% of the cases diagnosed in Western countries are currently classified as sporadic in addition to lacking an identifiable risk factor, thus limiting the rate of early detection to a better outcome [1,6].

### 2.1. Chronic Biliary Diseases

Primary sclerosing cholangitis (PSC), an autoimmune disorder which affects the bile ducts causing inflammation and obstruction, cholestasis, and progressive liver failure, represents a well-established risk factor for the development of CCA, especially in Western countries, and a 400-fold increase in the risk for CCA in patients diagnosed with PSC compared to the general population was reported [7,8,9]. The association of other inflammatory conditions, such as inflammatory bowel disease, further increases this risk. It is recommended that the patients diagnosed with PSC should be included in surveillance programs and lifelong follow-up protocols. Hence, they should be evaluated at least annually by imaging techniques and CA 19-9 (Carbohydrate antigen 19-9) serum assays [1,2,9,10,11,12].

Bile ducts cysts or choledochal cysts, representing a rare congenital disorder characterized by the cystic dilation of the intrahepatic and/or extrahepatic bile ducts, are more frequently diagnosed in Asian female patients who typically develop CCA much earlier than the general population (at a median age of approximately 32 years) [1,6,9]. The tumour may arise from the cysts as well as from the normal parts of the biliary tree. Bile stasis with increased concentration of bile acids, pancreatic enzymes reflux, and bacterial infection are some of the pathogenic mechanisms that are currently considered to be involved in the malignant transformation of the bile duct epithelium [1,9,13,14]. A recent meta-analysis concluded that bile duct cysts were the most important associated risk factor for all types of CCA [15].

Moreover, Caroli’s disease (a phenotype of the choledochal cyst) involves a genetic autosomal recessive disorder in which segmental intrahepatic bile ducts develop gross non-obstructive dilation [16]. This ailment is associated with a 38-fold higher risk for iCCA and a 97-fold higher risk for eCCA [17]. Under these circumstances, the tumour development mostly occurs after the second decade of life, although a few cases have been reported among teenagers [1,9,13,14,15,16,17]. Additionally, hepatholithiasis, also known as intrahepatic biliary lithiasis, represents another disorder associated with a higher risk of developing iCCA. This relationship has been well documented, especially in East Asia, where the disorder is more common than in Western countries, with an overall incidence of CCA between 5% and 13% [1,9,15,18]. Finally, cholelithiasis and choledocholithiasis have also been linked to an increased risk of CCA, particularly with eCCA, although a strong association with iCCA has also been suggested. The risk is thought to increase with gallstone size, the degree of calcification of the epithelium, and the duration of the disease [1,9,13,15,19].

### 2.2. Chronic Liver Conditions

Cirrhosis, a well-established risk factor for hepatocellular carcinoma, has also been identified as a risk factor for CCA. This aspect may be explained by the increased cellular proliferation, fibrosis, and release of pro-inflammatory cytokines that occur in the cirrhotic liver [1,9,15,20,21,22]. A meta-analysis of fourteen case–control studies, as well as a meta-analysis of seven case–control studies from 2012 and a population-based case–control study in Asian patients from 2013, identified cirrhosis as a strong risk factor for iCCA. Another study conducted in the US population concluded that cirrhosis might increase the risk for eCCA as well [20,21,22].

Chronic hepatitis B (HBV) and C (HVC) virus infections stand for another well-established risk factor for hepatocellular carcinoma which has been proven to increase the risk for CCA development too, particularly for iCCA [23]. In Western populations, iCCA was found to be more often associated with HCV infection, while in Asian populations, there is a stronger association was detected with HBV (which is endemic in Asia) [24]. Some data have also identified a small increase in eCCA risk in patients with chronic viral hepatitis, but this is still an open matter [24,25]. The cause of this increased risk seems to be not only the development of cirrhosis, which is a frequent complication of these viral infections, but also a direct carcinogenic effect of HBV and HCV on target cells [1,9,15,23,24,25,26,27].

In addition, hemochromatosis, a genetic disorder characterized by pathological iron accumulation in multiple organs, particularly in the liver, has been listed by some authors as risk factor for CCA, but this is still a matter of debate [28]. Some studies suggested an association with iCCA development, but no increased risk for eCCA was confirmed [29,30,31]. This higher risk may be explained by the co-presence of cirrhosis, a common clinical manifestation of hemochromatosis, but some iCCA cases have also been reported in non-cirrhotic hemochromatosis patients, thus suggesting supplementary pathogenic elements that are less understood so far [1,28,29,30,31,32,33]. Similarly, Wilson’s disease, another genetic disorder which affects the liver by pathological accumulation of copper, was pinpointed to present sporadic cases of iCCA according, for instance, to a cohort study conducted on 1186 patients (0.5% of who were confirmed with iCCA) [34,35,36,37]. An excessive copper accumulation may induce DNA (deoxyribonucleic acid) damage through reactive oxygen species (ROS) generation, yet some evidence has also shown a protective role of copper (at certain levels) against other different types of malignancies [1,34,35,36,37].

### 2.3. Digestive Ailments

Inflammatory bowel disease, specifically ulcerative colitis, has been found to be associated with a higher risk for iCCA development compared to Crohn’s disease, while the risk for eCCA was identified as similar between ulcerative colitis and Crohn’s disease [38]. Both conditions may lead to CCA by inducing a state of chronic inflammation, ROS anomalies, and/or microbiome dysbiosis [39]. As mentioned before, these conditions are also associated with PSC, which otherwise stands as a well-known factor for CCA [1,15,38,39,40,41]. Chronic pancreatitis, as well as duodenal or gastric ulcer, is still incompletely understood in relationship with the field of CCA, but it has been linked to a higher risk for eCCA than iCCA according to some data (yet there are currently no unanimous results) [1,13,15]. The underlying mechanism may be the fact that the biliary strictures which appear in up to 23% of patients diagnosed with chronic pancreatitis might further lead to cholangitis and cholelithiasis, both being well-known risk factors for CCA [1,15]. Also, a modest association between CCA diagnosis and duodenal gastric ulcer was highlighted in *Helicobacter pylori*-positive subjects [1,13,15].

### 2.4. Parasitic Infections

*Opisthorchis viverrini* and *Clonorchis sinensis*, also known as liver flukes, are two species of trematodes which have been identified as major contributors for CCA in some regions of the world [42]. Infection with these parasites is endemic in Eastern Asian countries and occurs through consumption of raw or undercooked fish [1,6,9,15,42,43,44]. The majority of CCA cases in these regions are linked to liver fluke infections, which tend to become chronic, with multiple reinfections despite an adequate anti-helminthic treatment [42,43]. This has led to both species to be now classified as group 1 biological carcinogens [42,43,44]. The parasites colonize the bile ducts, thus causing chronic inflammation, cholangitis, and fibrosis of the periportal system. In these cases, CCA might develop anywhere along the biliary tree, and it presents as any of the three anatomical subsets, although a higher incidence of iCCA has been reported [1,6,9,15,43,44,45,46,47,48,49].

### 2.5. Lifestyle Influence

The association between chronic alcohol consumption and CCAs has not been clearly confirmed, but some studies reported an increased risk for the development of iCCA [1,6,9,15,44,45]. However, it is less understood if the association is related to alcohol-induced liver disease or to other carcinogenic mechanisms [44,45]. With regard to eCCA, a similar risk between drinkers and non-drinkers was found that may be explained by alcohol-induced inhibition of gallstone formation via its inhibition of cholesterol metabolism [9,15,50,51,52,53,54]. With concern to (traditional) cigarette smoking, older studies have suggested that smoking may display a carcinogenic effect on the biliary epithelium because of the carcinogenic compounds that are metabolized by the liver and excreted in the bile [55,56,57]. More recently, a positive association between smoking and iCCA, as well as eCCA, has been reported [55,56,57,58]. Currently, no specific data have been reported with regard to the long-term use of electronic cigarettes and electronic nicotine delivery systems (ENDSs) with/without (concurrent or prior) conventional tobacco smoking and CCAs-related risk, but recent evidence suggested the involvement of the heavy metals that are found in the refill liquids in certain types of carcinogenesis [58,59,60].

### 2.6. Environmental Exposure

Thorotrast, a radiographic contrast agent that had been used during the first half of the 20th century, is currently banned because of its emission of alpha-radiation and subsequent carcinogenic effect [61,62]. The risk of CCA has been reported to be three hundred times higher in people exposed to this substance [61,62,63]. Regarding asbestos, several studies have found a link between asbestos exposure and CCA [64,65,66], results which have been confirmed by a case–control population-based study on the Nordic Occupational Cancer cohort, whereas a cumulative exposure to asbestos has been shown to increase the risk of iCCA but not eCCA [66].

### 2.7. Genetic and Epigenetic (Potential) Interplay

Certain genetic polymorphisms are associated with an increased risk for CCA. Some of the genes identified so far encode enzymes that are involved in xenobiotic detoxification, immune response, multidrug resistance, DNA repair, and folate metabolism [1,6,67,68,69,70]. Such genes vary from *BRCA* to *TBX3* according to research ranging from small sample-sized clinical studies to human cell experimental models, with this issue currently representing an emergent topic to be further studied [69,70]. Recently, gut microbiota and associated metabolomics analysis and RNA (ribonucleic acid) profiling showed that glutamine-associated signal transduction pathways might play a pivotal role in the development of iCCA, while experimental evidence suggested that glutamine supplementation may inhibit ferroptosis and downregulate *NOX1* (KNOTTED-like homebox) expression, which further inhibits NOX (NADPH oxidase) and ROS formation, as well as oncogene-related proteins such as p53 [67,71].

## 3. Metabolic and Endocrine Interferences in CCA Development

### 3.1. Non-Alcoholic Fatty Liver Disease (NAFLD)

NAFLD, the accumulation of lipids in the liver in the absence of alcohol abuse, ranging from steatosis to non-alcoholic steatohepatitis (NASH) and cirrhosis, is a very frequent ailment nowadays, associated or not with other endocrine, metabolic, and even cardiovascular traits [72,73,74,75]. This metabolic disorder has been linked to hepatocellular carcinoma diagnosis, but its involvement in the pathogenic features of CCAs is less clear according to the current knowledge. A US population-based study reported a positive association between NAFLD and iCCA, while a case–control study conducted in Japan identified NASH as an independent risk factor for iCCA [76,77,78,79]. These results were confirmed by a meta-analysis of seven case–control studies which concluded that NAFLD has a stronger association with iCCA rather than eCCA [78]. However, a recent study which included 180 patients from Italy and France showed that NASH, but not NAFLD, increases the risk for iCCA and might affect its prognosis. In this study, NASH was found in >20% of patients without classical risk factors for CCA, and in 80% of the cases, the malignancy developed in the liver without severe fibrosis. Thus, to some extent, NASH might explain the increasing worldwide incidence of iCCA [79].

### 3.2. Obesity

A high body mass index, particularly in the obesity range, might increase the risk of certain cancers, including CCAs, through fat-related effects on leptin, adiponectin, and pro-inflammatory cytokines and various growth factors, but most studies consider that current evidence is too limited to draw any definitive conclusions [1,5,9,15,80,81,82,83]. However, a recent study based on data collected from eleven European countries reported that more than 50% of the patients were overweight or obese at the time of CCA diagnosis and almost 40% suffered from hypertension, both (obesity and high blood pressure) comorbidities being more frequent among patients diagnosed with iCCA [80]. Also, a meta-analysis of five cohort studies and five case–control studies reported a small, positive association between obesity and CCA [83].

### 3.3. Type 2 Diabetes Mellitus

An important part of the metabolic panel concerns the diagnosis of type 2 diabetes, which has been shown to increase the risk for all CCA subtypes, but especially for iCCA [84]. Diabetic patients treated with metformin had a lower risk for CCA compared to those not receiving this oral medication, suggesting a potential protective role of metformin [85]. Insulin, which has a compensatory hypersecretion in the early stages of type 2 diabetes (in addition to the insulin resistance at receptors levels), has been shown to potentially stimulate cancer cell growth (hence acting as a growth factor) [86]. Diabetes might also increase the risk of biliary stones that further represent a risk factor for eCCA [1,5,9,15,84,85,86].

Moreover, bile salts’ function as nutrient signalling hormones is to activate different G-protein coupled nuclear receptors such as FXR, PXR, or S1PR2 (sphingosine-1 phosphate receptor 2) and collaborate with insulin amid the regulation of nutrients metabolism at the hepatic level via the activation of AKT and ERK1/2 pathways [87,88,89]. Anomalies of these loops may cause NAFLD or even directly contribute to CCAs’ development (for instance, through activation of S1PR2) [88]. At the clinical level, we mention, for instance, a multicentre study from 2021 on 537 consecutive patients diagnosed with CCAs that pinpointed the potential anti-cancer benefits in individuals receiving metformin versus subjects who had never used it (hazard ratio of 0.7, *p* = 0.0162), the former of whom experienced an advanced stage correlated with a better overall survival as well as those who had a sequential therapy in terms of chemotherapy followed by metformin versus chemotherapy (metformin-free) (hazard ratio of 0.44, *p* = 0.0016) [89].

### 3.4. Vitamin D Status

Vitamin D metabolism has been related to different malignancies in terms of vitamin D deficiency and a higher proliferative risk on one hand, or, on the other hand, the modulation of vitamin D receptors (and variations of different polymorphisms) that are displayed all over the human body, some of which might be prone to cancer development [90,91]. However, interventional trials in terms of cancer prevention or improving a poor neoplasia-related outcome are less clear so far [90,91,92]. The above-mentioned analysis also showed a longer disease free survival in subjects who underwent vitamin D supplementation when compared to those who were never users (hazard ratio of 0.55, *p* = 0.02) [89]. Of note, vitamin D might act as an anti-tumour drug via its non-mineral effects, but the exact regime and time frame of exposure as well as the potential interferences in terms of efficacy and toxicity of standard chemotherapy still represent an open issue [93,94,95]. For instance, CCA cell lines and animal studies have shown that 1-alpha, 25-dihydroxyvitamin D3 (the active form of vitamin D) analogues such as MART-10 (which enhances the role of calcitriol by 10 times) mediate anti-growth functions in human CCAs cells via the generation of cell cycle arrest [96]. Moreover, a higher vitamin D receptor expression (as revealed by the immunohistochemistry analysis) in CCA specimens from humans was associated with a better prognosis [97]. Further clinical studies are necessary to provide the exact level of evidence with respect to the vitamin D applications amid daily practice in this devastating malignancy [96,97]. Most probably, this loop represents one of the multiple other pathogenic traits of the condition.

### 3.5. Glucagon-like Peptide 1 Receptor (GLP-1R)

Novel insights of emergent anti-diabetic drugs such as GLP-1R agonists have suggested that they might act as anti-CCA agents, but the results are not convincing yet. On one side, an immunohistochemistry study published in 2024 showed that iCCA tissue displayed a statistically significant correlation (*p* = 0.027) between GLP-1R expression and a poor histological grading [98]. On the other side, a candidate to GLP-1R agonists such as liraglutide or exendin-4 might show a clinical improvement since there are in vitro experiments confirming that liraglutide (not exendin-4) reduced the CCA cells migration by interacting with epithelial–mesenchymal transition [98]. In vivo (murine) data revealed that liraglutide-related GLP-1R downregulation reduced the tumour volume through suppression of Akt/STAT3 signal transduction pathways [98]. Noting these apparently paradoxical effects on CCA, other clinical studies raised the issue of incretins (such as dipeptidyl peptidase 4 inhibitors (DDP4)) and GLP-1R agonists in increasing the risk of CCA [99]. For instance, a nationwide, registry-based cohort in three European countries (Sweden, Denmark, and Norway) analysed data between 2007 and 2018 and concluded there was no increase in the CCA risk amid the use of DPP4 and GLP-1R agonist when compared to the use of sulfonylureas [99].

### 3.6. Galanin System

This system includes the galanin peptide and its receptors GAL1-3-R (which are G protein-coupled receptors) that have been studied in relationship with various tumours, and lately, some immunohistochemistry (antibodies-based) analyses identified GAL1-R in bile duct cells and GAL3-R in cholangiocytes and capillary vessels, while in pCCA, GAL3-R expression may work as a poor prognosis marker according to small-sample-sized studies [100,101]. Since GAL1-3-Rs are confirmed (but with different configurations) in both healthy humans and CCAs (particularly, pCCA), further experimental models are necessary to highlight their significance as prognostic biomarkers [100,101]. Similarly, the system of this neuropeptide of 30 amino acids has been studied in relationship with other endocrine and non-endocrine tumorigenesis such as glioma, meniningioma, and pituitary neuroendocrine neoplasia [102]. One of the most interesting mechanisms related to the development of tumours stands from the observation that GAL-Rs were identified at the level of tumour-related immune cells that might play an important role in the tumour microenvironment homeostasis, as seen with other peptides including amid the metastatic process [102,103,104,105].

### 3.7. Sex Hormone Therapy

Both oestrogen (alpha and beta) and androgen receptors are expressed in the liver and biliary duct cells; thus, the potential interferences with neoplasia originating at this level should be taken into account [106]. In patients confirmed with CCAs, prior or concurrent administration of oestrogens (associated or not with progestins) in terms of oral contraceptive or menopausal-related hormone replacement, respectively, or of testosterone (which was offered for different purposes such as hypogonadism-related substitution or gender-affirming hormone therapy), represents a very challenging, cross-disciplinary, and modern perspective that, to our understanding, includes at least three main turning points, as follows:

Firstly, there is the big picture of oestrogens use for women of reproductive age or early post-menopausal females, noting that oestrogens-associated signal transduction pathways are involved in the underlying pro-carcinogenetic mechanisms of different tumours, not only at the biliary tract level [106,107]. Combined oral contraceptives, since their introduction in the early 1960s, have been proven to associate a mixed panel of neoplasia interplay, while generally being regarded as safe and effective. Recent data have shown that the overall cancer risk might not be influenced in current and prior users [107]. A transitory risk in current/regular users has been reported for mammary cancer (with a 20–30% risk increase). A risk decrease was found with regard to endometrial, colorectal, and ovarian malignancies (apart from those subjects coming from high-risk families as seen in the carriers of different pathogenic variants such as *BRCA*, *PTEN*, etc.). Notably, the risk is related to the timeframe of administration; for instance, a higher risk of cervical cancer was reported in women who were exposed for more than 5 years, a risk that is no longer active when the medication is stopped. Similarly, conflicting data are described in relation to the confirmation of hepatic adenocarcinomas, adenomas, and CCAs, with some studies suggesting a higher risk of iCCA in certain subgroups [107,108,109]. The identification of oestrogen receptor 1 configuration as a key factor in iCCA recurrence might impact the prognosis by interfering with the JAK/STAT3 pathway [110].

Moreover, the expression of CYP19A1, the aromatase that converts androgens to oestrogens, was found to be high in association with elevated oestrogen-related proteins within CCA tissues that displayed a more aggressive outcome, specifically in males rather than in females, and further exploration of this topic is required [111,112]. With concern related to menopause hormone therapy, we mention a nested case–control study from the UK from 2022 showing that oestrogen-only formulas decreased the risk of CCA (odd ratio of 0.59, 95% CI between 0.34 and 0.93), while the dose, duration, and time since last prescription were not found to be associated with CCA risk [113]. Interestingly, data from the Liver Cancer Pooling Project and the UK Biobank pinpointed that a history of hysterectomy was associated with a two-time increase in iCCA risk (hazard ratio of 1.98, 95% CI between 1.27 and 3.09) compared to females with physiological menopause who were aged between 50 and 54 years. Of note, more than 9 years of oral contraceptive exposure induced a 62% increase in iCCA risk (hazard ratio of 1.62, 95% CI between 1.03 and 2.55) [114].

Secondarily, the use of chronic androgen therapy, including anabolic androgenic steroids, poses a low level of statistical evidence with respect to the potential risk elevation of developing CCA according to the data we have so far [115].

Thirdly, the recently developed gender-affirming hormone therapy for transgender males is under evaluation with concern to the risk of hepatobiliary neoplasms, but no clear conclusion has been established yet. The clinical data remain at isolated case reports [116]. An expansion of the published data in this particular multidisciplinary area is expected within the following years (Table 1).

## 4. Paraneoplastic Syndrome in CCAs

CCAs may cause a variety of paraneoplastic manifestations underlying various endocrine and non-endocrine mechanisms which are less or more understood at the present time. Recognition of these syndromes might lead to an early diagnosis of CCA, while their presence may predict the efficacy of treatment, including a relapse or recurrence of the disease. Although the available data on this specific subject are rather limited when compared to the malignancies of other origins, there are reports of dermatological, neurological, haematological, renal, and endocrine manifestations in CCAs. Of note, the current level of evidence remains mostly low, but their awareness is essential for daily basis [3,117].

### 4.1. Dermatological Features Have Been Found as Followings

#### 4.1.1. Acanthosis

Acanthosis nigricans was associated with various solid malignancies including CCA. It presents as hyperkeratotic hyper-pigmented diffuse patches on the palms and soles, also affecting mucosa such as the oral cavity and oesophagus [3,118].

#### 4.1.2. Alopecia

Alopecia areata is usually the type of hair loss associated with CCA (while not being specific), presenting as a round patch with well-shaped edges on top of the skull [119,120,121].

#### 4.1.3. Dermatomyositis

Dermatomyositis represents a rare paraneoplastic manifestation of CCA, resulting from autoimmune response against the cancer cells which cross-reacts with healthy tissues. It is characterized by weakness of the proximal muscles, heliotrope rash (blue-purple skin lesions on the upper eyelids), and Gottron papules (erythematous papules on knuckles) [122,123].

#### 4.1.4. Porokeratosis

Disseminated superficial porokeratosis is regarded as a benign hyperkeratotic skin tumour which has been linked to various malignancies. It displays itchy reddish-brown papules with elevated borders on the extensor surface of the limbs and trunk, sparing mucosal surfaces, palms, and soles. They become visible a few months before cancer-related symptoms appear; thus, prompt recognition and further investigations might help the overall outcome [124].

#### 4.1.5. Necrotic Migratory Erythema

This lesion has been found in association with CCA too. It presents as annular erythematous erosive lesions of 1 to 2 cm that are surrounded by scaling, localized on the face, trunk, and extremities [125].

#### 4.1.6. Persistent Erythema Multiform

This skin lesion is a disorder classified as paraneoplastic dermatosis because it appears in the setting of internal organ malignancies. In CCA, it usually occurs early in the course of the disease and presents as a painful erythematous rash with scaling, forming patches on the upper trunk and thighs; after a few days, it evolves into haemorrhagic bullae with violaceous edges which do not heal on their own; vital signs are normal and mucosal surfaces are not involved, as seen in other malignancies or as side effects to different drugs [126,127,128].

#### 4.1.7. Sweet Syndrome

This ailment highlights an acute febrile neutrophilic dermatosis, occurring in approximately 15% of the people suffering from solid malignant tumours. It is characterized by rapidly growing painful erythematous plaques on the face, neck, and legs, associated with malaise, cough, and arthralgia. CCA and other malignancies involve an excessive production of granulocyte-colony stimulation factor (G-CSF) which stimulates neutrophils, causing this clinical presentation [129,130,131].

#### 4.1.8. Bazex Syndrome

Also known as acrokeratosis paraneoplastica, this is another syndrome reported to be associated with CCA. It presents as hyperkeratotic pruritic scaly lesions of the face, ears, buttocks, palms, and soles, which can be associated with fatigue, abdominal pain, nausea, vomiting, constipation, and weight loss [132].

#### 4.1.9. Erythema

Erythema giratum reflects a migratory erythema most commonly associated with oesophageal, lung, and breast carcinoma, but it has also been reported in association with CCA [133,134,135,136,137].

#### 4.1.10. Pityriasis

Pityriasis rubra pilaris represents another dermatosis reported to be associated with CCA, presenting as a papulosquamous pruritic rash associated with mild fever [138].

#### 4.1.11. Lupus

Subacute cutaneous lupus erythematosus, a rare paraneoplastic manifestation of CCA, involves a clinical presentation showing an explosive onset of a pruritic rash, with arthralgia associated with a lower limb oedema [139].

#### 4.1.12. Leser–Trelat Sign

It is defined as the presence of multiple pigmented seborrheic keratosis associated with an underlying malignancy and was mostly reported in gastrointestinal adenocarcinoma and rarely with CCA. In association with acanthosis nigricans, it stands for an even stronger indicator of a malignant disease [140,141].

#### 4.1.13. Porphyria

Porphyria cutanea tarda, the most common type of porphyria, resulting from a deficient activity of uroporphyrinogen decarboxylase, an enzyme involved in hem biosynthesis, poses clinical manifestations due to an increased mechanical fragility of the skin after sunlight exposure, leading to erosions and blistering, painful sores, milia, depigmentation, and scarring. Excretion of porphyrins in urine and stool is characteristically increased. There are many reports of porphyria in the setting of hepatocellular carcinoma, but in the case of CCA, porphyria may be considered an exceptional occurrence [3,142].

### 4.2. Neurological Paraneoplastic Elements 

Neurological paraneoplastic elements are rare findings in cases diagnosed with CCAs, but limbic encephalopathy and paraneoplastic cerebellar degeneration have been reported. Limbic encephalopathy is a subacute cause of memory impairment and confusion characterized by autonomic seizures (pilomotor erection), delusion, and rapidly progressive dementia. Paraneoplastic cerebellar degeneration presents as sudden onset ataxia, dysarthria, diplopia, and dysphagia in a patient with underlying malignancy [143,144].

### 4.3. Renal Findings

Renal findings are extremely rare; a case of fibrillary glomerulonephritis has been reported in association with CCA; the clinical presentation was with oedema of the lower limbs and face, uncontrolled hypertension, nephrotic range proteinuria, and microscopic haematuria [145].

### 4.4. Haematological Manifestations

Haematological manifestations include a heterogeneous spectrum. For instance, paraneoplastic vasculitis represents less than 5% of all vasculitis cases and it is more commonly associated with haematological malignancies than with solid tumours. Most cases present as cutaneous forms, but internal organs may also be involved too. Anti-neutrophilic cytoplasmic antibodies (ANCA) titre is usually raised [146,147]. Giant cell arteritis has been reported to be more commonly associated with CCAs, but one study also reported polyarteritis nodosa (PAN) to be associated with this neoplasia. Early signs of PAN are bilateral numbness of lower limbs and gradually increasing fever. After two weeks, an arthritis-like disorder develops, with severe pain affecting the ankles, metatarsal, and phalangeal joints; skin involvement with necrosis and gangrene on distal phalanges is reported to develop after a few months; in late stages, severe abdominal pain, nausea, and vomiting appear as part of the gastrointestinal tract involvement. Paraneoplastic PAN is poorly responsive to steroids, with tumour removal or chemotherapy being needed for resolution of vasculitis [146,147].

Moreover, Trousseau’s syndrome, migratory superficial phlebitis, was identified as a sign of malignancy for the first time in 1865 by the French physician Armand Trousseau [147,148,149]. More than 100 years later, the definition of original syndrome was expanded to include any venous or arterial thromboembolism occurring in the presence of an underlying malignancy, including chronic disseminated intravascular coagulopathy, micro-angiopathic haemolytic anaemia, and Libman–Sacks endocarditis, while currently, the syndrome is defined as the occurrence of unexplained thrombotic events before or simultaneously with the diagnosis of a visceral malignancy such as those originating from the brain, lung, ovaries, and gastrointestinal tract, including hepatocellular carcinoma, with association with CCA being less common [148,149]. Some data showed that 10% of subjects with confirmed CCAs developed venous thromboembolism, which was associated with a reduced survival rate. It was hypothesized that the cause of Trousseau’s syndrome is hypoxia leading to activation of endothelial adhesion molecules and the coagulation pathways. Low-molecular-weight heparin and removal of the primary tumour have been found to be the most effective treatment strategies. On the other hand, using anticoagulants to treat venous thrombosis and prevent pulmonary thromboembolism while needing to perform a diagnostic liver biopsy can be challenging [148,150]. Additionally, antiphospholipid antibody syndrome, a cause of hypercoagulability due to formation of antibodies against platelet phospholipids, has been linked to various solid organ malignancies, including CCA [151,152]. We also mention the paraneoplastic leukemoid reaction, a manifestation defined by a number of white blood cells of more than 50,000/mm^3^, particularly neutrophils, in reaction to any infection or carcinoma. In the setting of CCA, it has been reported to mimic a pyogenic liver abscess, presenting as intermittent fever for at least a month, progressive generalized weakness, weight loss, and leucocytosis [153,154,155].

Of note, multi-systemic paraneoplastic syndrome might also include adult-onset Still disease, which was found in one study in relationship with CCA; clinical presentation included high-grade fever and chills, sore throat, cough, myalgia, pleuritic chest pain, and arthralgia involving various joints [156]. Newly diagnosed systemic lupus erythematosus with full blown clinical picture was reported as a paraneoplastic presentation in CCA too [157,158].

### 4.5. Humoral Manifestations

Humoral manifestations are mostly dominated by the diagnosis of hypercalcaemia of malignancy that overall had been reported in some CCA cases. Most of them are related to the over-production of parathyroid hormone-related protein (PTHrP), also named parathyroid hormone-like hormone (PTHLH), by the proliferating bile duct epithelial cells. PTHrP causes an elevation of the serum calcium via interacting with tumour growth factor alpha (TGF-α) and tumour necrosis factor alpha (TNF-α), with PTH levels being normal or suppressed by the blood calcium elevation [84,159]. The usual clinical signs are caused by the increased serum calcium, specifically abdominal discomfort, constipation, nausea, bone pain, polydipsia, polyuria, alteration of the mental status (even syncope [160]), and weakness [159,160,161,162,163].

As mentioned, the tumour-related secretion of G-CSF might serve as a paraneoplastic element as well as a biomarker of disease progress or recurrence. In 2017, a third-ever case of co-secretion with concern to CCA-related G-CSF and PTHrP was reported in a 78-year-old male who was initially suspected to have a liver abscess. After the histological diagnosis was confirmed via percutaneous needle biopsy, the subject started chemotherapy and died 134 days after its initiation, hence highlighting the importance of early ailment recognition and the fact that a humoral paraneoplastic syndrome usually indicates a severe outcome [161].

Additionally, we mention another interesting tumour biology that triggers hypercalcaemia: the presence of a fibroblast growth factor receptor-positive iCCA (the most recent report was provided in 2024) [162]. In 2020, a first-ever case of CCAs, specifically iCCA, was reported to associate humoral hypercalcaemia of malignancy with polycythaemia and leucocytosis, respectively, with elevated procalcitonin (in the absence of any infection) in an 80-year-old man [163] (Figure 1).

## 5. Discussion

CCAs, very aggressive neoplasia, need to be integrated amid the modern medicine era, including the understanding of the particular features with respect to the tumour biology, the constellation of the risk factors, and the clinical picture, from the first presentation and the co-presence of paraneoplastic syndromes to the use of various imagery tools across an overall multidisciplinary management. This group of ailments still presents a large area of pitfalls and challenges despite recent progress in imagery assessment, biomarkers, surgical techniques, and chemotherapy/immunotherapy, and even liver transplantation in certain centres [164].

### 5.1. Imagery Tools to Help the Clinical Assessment

An early diagnosis might improve the overall prognosis, and this hinges on the clinical evaluation as well as the imagery investigations in addition to the blood assays. Transabdominal ultrasound (TAUS) is often the first and most accessible imaging technique for the evaluation of patients with right upper abdominal pain with or without jaundice because it detects bile duct dilatation, the presence of gallstones, or other causes of biliary obstruction as well as other hepatic masses. In CCAs, depending on the anatomical type, TAUS might show bilateral intrahepatic duct non-union and segmental dilatation or polypoid masses inside the biliary tract representing papillary tumours or smooth masses with mural thickening in nodular pCCA. The intrahepatic form of CCA might present a large hepatic mass with irregular margins; depending on the amount of fibrosis, mucin, and calcification within the tumour, the echogenicity can be predominately hypo- or hyperechoic or mixed. The distal common bile duct may be often masked by the duodenal air; therefore, TAUS does not identify a tumour at this level. However, distention of the bile ducts is used as an indicator of distal biliary obstruction [165,166].

Contrast-enhanced ultrasound (CEUS) has been successfully used to characterize focal liver lesions based on specific enhancement and washout patterns, having a superior diagnostic ability compared to grey scale and Doppler ultrasound. CEUS might misdiagnose iCCA as hepatocarcinoma [167]. Computed tomography is applied in up to 90% of suspected CCAs due to its ability to assess tumour extension and potential resectability, as well as vascular involvement and lymph node metastases. In patients with eCCA, the tumour site is suggested by the level of biliary dilatation. Dilatation of both intrahepatic ducts and separation of the left and right hepatic ducts is suggestive for pCCA. Magnetic resonance imagery also provides a very good assessment of tumour extension, vascular involvement, and resectability. These imaging techniques also have an important role in the surveillance of patients with an increased CCA risk, such as those with PSC [167,168].

### 5.2. Is There a Place for Endocrine Considerations in CCAs?

Nowadays, one of the most challenging sub-domains in the field of CCAs remains the potential link with the endocrine traits as tumour-related manifestations. An in-depth perspective of the hormonal paraneoplastic profile in CCAs showed that it remains at a very low level of statistical evidence; thus showing the importance of its awareness amid a multidisciplinary complex perspective. According to the data we have so far in CCAs, humoral hypercalcaemia of malignancy stands for the most frequent humoral paraneoplastic syndrome in CCAs, despite being exceptional amid the identification of other paraneoplastic (non-endocrine) manifestations and compared to its reported frequency in other (non-CCAs) cancers (it accompanies 20–30% of malignancies) [169,170]. It represents a poor prognosis marker in CCA as seen in other tumours (except for the majority of mammary cancers, where it might imply a long standing evolution). It may be episodic once the tumour relapses. In addition to the therapy that targets the originating malignancy, the control of elevated serum calcium requires the administration of a bisphosphonate such as intravenous zoledronic acid or subcutaneous denosumab, a human monoclonal antibody which acts as an inhibitor of RANKL (receptor activator of nuclear factor kappa B ligand), as similarly used in PTH-related hypercalcaemia (e.g., primary hyperparathyroidism) [171]. Early detection firstly helps the general wellbeing of a patient due to prompt medical control of hypercalcaemia and also provides a fine biomarker of disease status in selected cases that harbour the capacity of PTHrP secretion. The exact molecular biology and genetic configuration of CCAs that display such characteristics is still an open matter, but humoral hypercalcaemia adds to the overall disease burden [160,161,162,163,172].

Generally, the underlying mechanisms in humoral hypercalcaemia of malignancy are associated with PTHrP (PTHLH) over-secretion as described in CCAs which might act as growth promotor via activation of the ERK (extracellular signal-related kinase)–JNK (c-Jun N-terminal kinase)-ATF2 (activating transcription factor-2)-cyclinD1 pathway (this canonical path explains the paracrine effects of PTHrP) [173]. Other (non-CCAs) tumour biologies that also impair the blood calcium are represented by the ectopic PTH over-secretion (at the level of malignant cells, not by the parathyroid glands) or abnormal production of 1,25-dihydroxyvitamin D within malignancies (particularly hematologic neoplasia such as lymphomas, as similarly found in sarcoidosis and granulomatous diseases) that hold the enzymatic equipment for this final step of 25-hydroxyvitamin D activation (calcitriol-induced hypercalcaemia). In addition, the presence of bone (osteolytic) metastases might increase the serum calcium by displacing a large bone mass [174,175,176]. (Figure 2).

### 5.3. Current Limits and Further Expansion

The level of endocrine evidence across clinical trials remains far from generous in CCAs; thus, we performed a narrative analysis rather than a systematic review. We need more experimental models to understand the tumour biology with respect to the hormonal interplay, and further applications such as emergent biomarkers or new strategies are expected for daily practice. Other topics in this specific area that refer to the hormonal insights in CCAs (which remain out of the scope of the present work) might include thyroid hormones’ implications in the liver and biliary duct tract under physiological and pathological circumstances (such as goitre or thyroid cancer) or the analysis of the CCA subgroups that displays adrenal metastases complicated with adrenal insufficiency [177,178,179]. Moreover, the recent COVID-19 pandemic expanded the edges of the scientific knowledge as well as identifying new clinical/surgical entities; novel research has delved into system biology approaches to connect the viral infection with a higher risk of CCAs or to prove whether CCAs patients were prone to developing a more severe respiratory panel amid coronavirus complications [180,181,182,183,184,185].

This works brings novel information to bridge the gap toward the traditional data in CCAs that helps in understanding the tumour behaviour with respect to the clinical implications in terms of the paraneoplastic syndromes and with regard to a large panel of potential contributors that come from different medical areas. To our knowledge, a particular endocrine frame as a novel approach across this complex picture represents an emergent topic to further be studied and understood.

## 6. Conclusions

The hormonal burden in CCAs is reflected in a complex constellation of interplays such as the galanin system, exposure to sex hormone therapy, interferences with vitamin D system, etc., while humoral hypercalcaemia of malignancy, despite rare, has been reported due to PTHrP over-production in addition with other elements such a G-CSF, fibroblast growth factor receptor positivity, or even procalcitonin. The level of endocrine evidence across clinical trials remains far from generous in CCAs. Further applications such as emergent hormonal biomarkers or new strategies are expected for daily practice.

## Figures and Tables

**Figure 1 biology-13-00662-f001:**
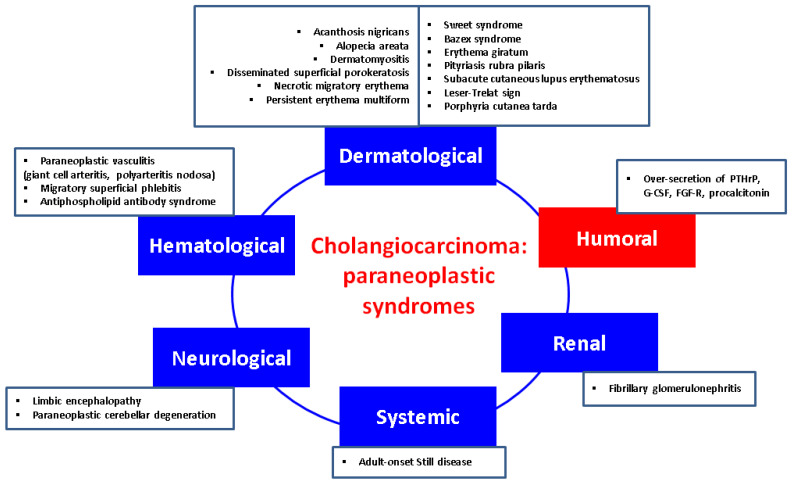
Integrating humoral (endocrine) paraneoplastic syndrome amid the complex multidisciplinary panel of paraneoplastic elements in CCAs: a qualitative perspective [117,118,119,120,121,122,123,124,125,126,127,128,129,130,131,132,133,134,135,136,137,138,139,140,141,142,143,144,145,146,147,148,149,150,151,152,153,154,155,156,157,158,159,160,161,162,163]. (Abbreviations: FGF-R = fibroblast growth factor receptor; G-CSF = granulocyte-colony stimulation factor; PTHrP = parathyroid hormone-related protein).

**Figure 2 biology-13-00662-f002:**
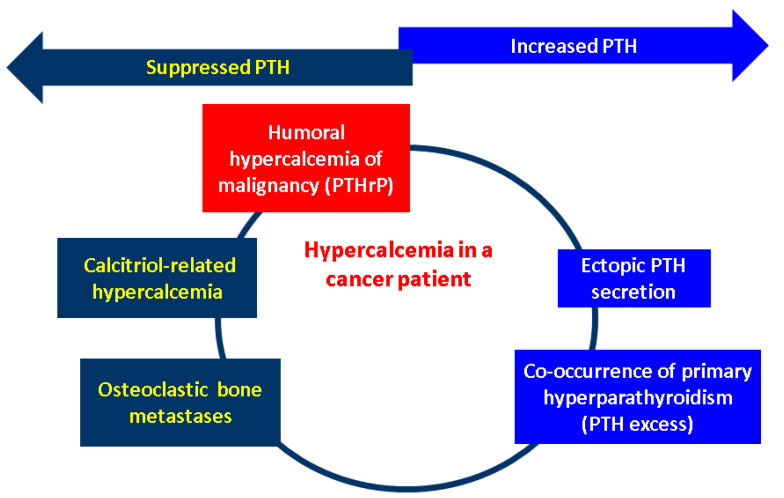
Mechanisms of hypercalcaemia in patients diagnosed with a malignancy: humoral hypercalcaemia of malignancy due to PTHrP excess; ectopic PTH secretion (by a non-parathyroid tumour); osteolytic metastases; calcitriol-induced hypercalcaemia; and potential co-occurrence of the primary hyperparathyroidism-associated hypercalcaemia due to an orthotopic or ectopic parathyroid gland [170,171,172,173,174,175,176]. High calcium levels are induced by the PTHrP excess, which causes the suppression of the physiological PTH at the level of parathyroid glands via negative feedback. Calcitriol over-production also causes hypercalcaemia by interfering with the metabolism of vitamin D, and this also supresses the normal parathyroid-related PTH. Osteolytic metastases displace a large bone mass, and this comes with the serum release of the calcium, thus causing physiological PTH inhibition. One the other hand, a synchronous parathyroid tumour causes an abnormally high PTH that induces hypercalcaemia (primary hyperparathyroidism). The same phenomenon takes place in cases with ectopic PTH secretion (involving other cancers than parathyroid glands tumours). (Abbreviations: PTH = parathormone; PTHrP = PTH-related protein).

**Table 1 biology-13-00662-t001:** Risk factors and potential contributors in CCA development (a qualitative analysis).

Conditions	Key Elements	Reference Numbers
Chronic biliary diseases	Primary sclerosing cholangitis	[7,8,9]
	Bile ducts cysts or choledochal cysts (including Caroli’s disease)	[13,14,16,17]
	Hepatholithiasis	[17]
Chronic liver conditions	Cirrhosis	[1,9,15,20,21,22]
	Hepatitis B and C virus chronic infections	[15,23,24,25,26,27]
	Hemochromatosis	[1,28,29,30,31,32,33]
	Wilson’s disease	[34,35,36,37]
Digestive ailments	Inflammatory bowel disease (ulcerative colitis, Crohn’s disease) Chronic pancreatitis Duodenal or gastric ulcer	[1,13,15,38,39]
Parasitic infections (liver fluke)	*Opisthorchis viverrini* or *Clonorchis sinensis*	[1,6,9,15,42,43,44]
Lifestyle influence	Chronic alcohol consumption	[44,45]
	Cigarette smoking	[55,56,57]
Environmental exposure	Thorotrast	[61,62]
	Asbestos	[64,65,66]
Genetic and epigenetic considerations	*BRCA* to *TBX3,* p53	[69,70,71]
Metabolic and endocrine interferences	Non-alcoholic fatty liver disease	[78,79]
	Obesity	[80,81,82,83]
	Type 2 diabetes mellitus	[1,5,9,15,84,85,86]
	Vitamin D deficiency	[90,91]
	Modulation of glucagon-like peptide 1 receptor	[98]
	Modulation of galanin system	[100,101]
	Sex hormone therapy (oestrogens in adult females)	[107,108,109]

## Data Availability

Not applicable.

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
