# Peer review of "The Constellation of Risk Factors and Paraneoplastic Syndromes in Cholangiocarcinoma: Integrating the Endocrine Panel Amid Tumour-Related Biology (A Narrative Review)"

_biology, 2024, doi:10.3390/biology13090662_

Round 1

Reviewer 1 Report

Comments and Suggestions for Authors

The authors discuss recent updates on risk factors for cholangiocarcinoma, a common hepatic malignancy in this review. I have the following comments:

  1. The article lacks tables. Adding one or two tables would enhance clarity.
  2. Make the figures more detailed and improve the image quality.
  3. Several articles have been published on topics similar to your manuscript. Can you elaborate on the novelty of your research?
  4. The abstract should be more concise and clear.
  5. References do not follow the journal's formatting guidelines.
  6. Figure legends should be placed at the bottom of the figures, not at the top.
  7. Throughout the manuscript, references are cited at the end of paragraphs. This makes it unclear exactly where the information comes from. Please cite references immediately after the relevant information.
  8. Sentences in the manuscript are too long, making them difficult to understand (e.g., lines 391-396 make a single sentence)

Formatting errors:

  • Line 56: Do not use past tense.
  • Lines 307-308: The word "respectively" seems to be unnecessarily used.
  • Lines 526-539: Please provide reference.

Comments on the Quality of English Language

The manuscript requires thorough proofreading and English language correction.

Author Response

Response to Review 1 Comments

Dear Reviewer,

Thank you very much for your time and your effort to review our manuscript.

We are very grateful for providing your valuable feedback on the article.

Here is our response and related amendment that has been made in the manuscript according to your review (marked in yellow color).

The authors discuss recent updates on risk factors for cholangiocarcinoma, a common hepatic malignancy in this review.

Thank you very much.

I have the following comments:

The article lacks tables. Adding one or two tables would enhance clarity.

Thank you very much. We added Table 1.

Risk factors and potential contributors in CCA development (a qualitative analysis)

Conditions

Key  elements

Reference numbers

Chronic biliary diseases

Primary sclerosing cholangitis

[7,8,9]

Bile ducts cysts or choledochal cysts

(including Caroli’s disease)

[13,14,16,17]

Hepatholithiasis

[17]

Chronic liver conditions

Cirrhosis

[1,9,15,20-22]

Hepatitis B and C virus chronic infections

[15,23-27]

Hemochromatosis

[1,28-33]

Wilson’s disease

[34-37]

Digestive ailments

Inflammatory bowel disease

(ulcerative colitis, Crohn’s disease)

Chronic pancreatitis

Duodenal or gastric ulcer

[1,13,15,38,39]

Parasitic infections

(liver fluke)

Opisthorchis viverrini Clonorchis sinensis

[1,6,9,15,42-44]

Lifestyle influence

chronic alcohol consumption

[44,45]

cigarette smoking

[55-57]

Environmental exposure

Thorotrast

[61,62]

Asbestos

[64-66]

Genetic and epigenetic considerations

BRCA to TBX3, p53

[69,70,71]

Metabolic and endocrine interferences

Non-alcoholic fatty liver disease

[78,79]

obesity

[80-83]

Type 2 diabetes mellitus

[1,5,9,15,84-86]

Vitamin D deficiency

[90,91]

Modulation of Glucagon-like peptide 1 receptor

[98]

Modulation of Galanin system

[100,101]

Sex hormone therapy (oestrogens in adult females)

[107-109]

Make the figures more detailed and improve the image quality.

Thank you very much. We expanded Figure 1.

Figure 1. Integrating humoral (endocrine) paraneoplastic syndrome amid the complex multidisciplinary panel of paraneoplastic elements in CCAs: a qualitative perspective [117-163] (Abbreviations: FGF-R=fibroblast growth factor receptor; G-CSF=granulocyte-colony stimulation factor; PTHrP= parathyroid hormone-related protein)

We provided more explanations at Figure 2.

Figure 2. Mechanisms of the hypercalcemia in patients diagnosed with a malignancy: humoral hypercalcemia of malignancy due to PTHrP excess; ectopic PTH secretion (by a non-parathyroid tumour); osteolytic metastases; calcitriol-induced hypercalcemia, and potential co-occurrence of the primary hyperparathyroidism-associated hypercalcemia due to an orthotopic or ectopic parathyroid gland [170-176]. High calcium levels are induced by the PTHrP excess that causes the suppression of the physiological PTH at the level of parathyroid glands via negative feedback. Calcitriol over-production also causes hypercalcemia by interfering with the metabolism of vitamin D, and this also supresses the normal parathyroid-related PTH. Osteolytic metastases displace a large bone mass and this comes with the serum release of the calcium, thus causing physiological PTH inhibition. One the other hand, a synchronous parathyroid tumour causes an abnormally high PTH that induces hypercalcemia (primary hyperparathyroidism). The same phenomenon takes place in cases with ectopic PTH secretion (involving other cancers than parathyroid glands tumours).

Several articles have been published on topics similar to your manuscript.

Can you elaborate on the novelty of your research?

Thank you very much. In addition to the data that have been mentioned as controversial or ongoing studies at each sub-section, we added at Discussion: This works brings novel information to bridge the gap toward the traditional data in CCAs that helps the understanding of the tumour behaviour with respect to the clinical implications in terms of the paraneoplastic syndromes and with regard to a large panel of potential contributors that come from different medical areas. To our aware, a particular endocrine frame as a novel approach across this complex picture represents an emergent topic to further be studied and understood. 

The abstract should be more concise and clear.

Thank you very much. We corrected it.

References do not follow the journal's formatting guidelines.

Thank you very much. This is part of the final editing in agreement with the editorial team. Thank you

Figure legends should be placed at the bottom of the figures, not at the top.

Thank you very much. We corrected them.

Throughout the manuscript, references are cited at the end of paragraphs. This makes it unclear exactly where the information comes from. Please cite references immediately after the relevant information.

Thank you very much. We corrected them.

Sentences in the manuscript are too long, making them difficult to understand (e.g., lines 391-396 make a single sentence)

Thank you very much. We corrected them.

Formatting errors:

Line 56: Do not use past tense.

Thank you very much. We corrected it.

Lines 307-308: The word "respectively" seems to be unnecessarily used.

Thank you very much. We removed it.

Lines 526-539: Please provide reference.

Thank you very much. We corrected it.

Comments on the Quality of English Language: The manuscript requires thorough proofreading and English language correction.

Thank you very much. We corrected it.

Thank you very much.

Reviewer 2 Report

Comments and Suggestions for Authors

Overall, this review was written well. However, since this review paper was submitted to this journal of "Biology", authors should add more biology discussions in detail instead of clinical epidemiology, or very superficial and simple biology without deep understandings as currently laid out. It is recommended that authors may consider submitting this review paper to another MDPI journal "Biomedicines", where it fits better.

Minor: It seems the section 4.1 was written in a different style of using bullet points instead of narrative paragraphs. Please consider making this section look consistent with all other sections.

Comments on the Quality of English Language

Please double check the English grammars throughout the whole manuscript. The English of simple summary and abstract are difficult to understand. Please do a major revision.

Author Response

Response to Review 2 Comments

Dear Reviewer,

Thank you very much for your time and your effort to review our manuscript.

We are very grateful for your insightful comments and observations, also, for providing your valuable feedback on the article.

Here is a point-by-point response and related amendments that have been made in the manuscript according to your review (marked in yellow color).

Overall, this review was written well.

Thank you very much. We really appreciate it!

However, since this review paper was submitted to this journal of "Biology", authors should add more biology discussions in detail instead of clinical epidemiology, or very superficial and simple biology without deep understandings as currently laid out. It is recommended that authors may consider submitting this review paper to another MDPI journal "Biomedicines", where it fits better.

Thank you very much.

We extended the data at Discussion and we respectfully mention the followings:

  1. Tumor biology also means a certain pattern of clinical behavior as a direct consequence, as well as understanding the contributors, risk factors, triggers, and atypical expressions such as paraneoplastic syndromes.
  2. Due to the epidemiological aspects (for instance, the spreading of certain viral infections, the impact of obesity or vitamin D deficiency all over the world) the tumor biology might change and further one it might impact the clinical presentation, hence, we consider these aspects closely related one to another and be helpful for a multidisciplinary perspective. Moreover, all the pre-clinical data in humans, cell lines experiments, animal experiments are meant to serve for the benefit of the patients, thus a clinical translation of the studies that regard these aspects of the tumor biology are essential nowadays.
  3. Also, we quote from the invitation on the Guest Editors that stands as intro for this Special Issue entitled “Biology of Liver Diseases” whereas clinical aspects are specifically mentioned and welcomed according to the design of this special issue that manages an interesting, cross-disciplinary perspective from basic to clinical research.

“In this Special Issue entitled “Biology of Liver Diseases”, we invite a wide range of papers, including basic research, clinical research, and review articles.”

Thank you very much.

Minor: It seems the section 4.1 was written in a different style of using bullet points instead of narrative paragraphs. Please consider making this section look consistent with all other sections.

Thank you very much. The section has the same structure, but with smaller sub-sections. We separated them according to your recommendation. Thank you

Comments on the Quality of English Language: Please double check the English grammars throughout the whole manuscript. The English of simple summary and abstract are difficult to understand. Please do a major revision.

Thank you very much. We corrected the English language, the abstract, and the summary. Thank you

Thank you very much.

Reviewer 3 Report

Comments and Suggestions for Authors

In this manuscript, Mihai-Lucian Ciobica et al. have conducted a thorough review of recent clinical studies on Cholangiocarcinomas (CCAs), encompassing CCA-associated risk factors and potential contributors, metabolic and endocrine interferences in CCA development, and paraneoplastic syndrome in CCAs. The manuscript is of very high quality, offering a comprehensive overview of current clinical knowledge. Additionally, the discussion section provides valuable insights into disease biology and hypotheses. I have only one minor suggestion for the authors before the manuscript can be officially accepted and published in Biology: It would be beneficial if the authors could specify the time range of the literature search used in writing this review. If there was no specific range, please indicate the time range of most of the literature cited. This clarification will highlight the state-of-the-art nature of this review.

Author Response

Response to Review 3 Comments

Dear Reviewer,

Thank you very much for your time and your effort to review our manuscript.

We are very grateful for your insightful comments and observations, also, for providing your valuable feedback on the article.

Here is a point-by-point response and related amendments that have been made in the manuscript according to your review (marked in yellow color).

In this manuscript, Mihai-Lucian Ciobica et al. have conducted a thorough review of recent clinical studies on Cholangiocarcinomas (CCAs), encompassing CCA-associated risk factors and potential contributors, metabolic and endocrine interferences in CCA development, and paraneoplastic syndrome in CCAs. The manuscript is of very high quality, offering a comprehensive overview of current clinical knowledge. Additionally, the discussion section provides valuable insights into disease biology and hypotheses. 

Thank you very much. We really appreciate it!

I have only one minor suggestion for the authors before the manuscript can be officially accepted and published in Biology: It would be beneficial if the authors could specify the time range of the literature search used in writing this review. If there was no specific range, please indicate the time range of most of the literature cited. This clarification will highlight the state-of-the-art nature of this review.

Thank you very much. The search was done from Inception until June 2024 and most papers were published within latest decade and we added this information according to your recommendation. Thank you

Round 2

Reviewer 1 Report

Comments and Suggestions for Authors

I am satisfied with the revisions made by the authors to the manuscript.